# Assessing Cachexia Acutely after Autologous Stem Cell Transplant

**DOI:** 10.3390/cancers11091300

**Published:** 2019-09-04

**Authors:** Lindsey J. Anderson, Chelsea Yin, Raul Burciaga, Jonathan Lee, Stephanie Crabtree, Dorota Migula, Kelsey Geiss-Wessel, Haiming M. Liu, Solomon A. Graf, Thomas R. Chauncey, Jose M. Garcia

**Affiliations:** 1Geriatric Research, Education and Clinical Center, Veterans Affairs Puget Sound Health Care System, Seattle, 98108 WA, USA; 2Bone Marrow Transplant Unit, Veterans Affairs Puget Sound Health Care System, Seattle, 98108 WA, USA; 3Gerontology and Geriatric Medicine, University of Washington Department of Medicine, Seattle, 98195 WA, USA; 4Oncology, University of Washington Department of Medicine, Seattle, 98195 WA, USA; 5Fred Hutchinson Cancer Research Center, Seattle, 98109 WA, USA

**Keywords:** muscle wasting, muscle function, testosterone, hematopoietic stem cell transplant

## Abstract

Autologous hematopoietic stem cell transplantation (AHCT) is an accepted strategy for various hematologic malignancies that can lead to functional impairment, fatigue, muscle wasting, and reduced quality of life (QOL). In cancer cachexia, these symptoms are associated with inflammation, hypermetabolism, and decreased anabolic hormones. The relative significance of these factors soon after AHCT setting is unclear. The purpose of this study was to characterize the acute effects of AHCT on physical function, body composition, QOL, energy expenditure, cytokines, and testosterone. Outcomes were assessed before (PRE) and 30 ± 10 days after (FU) AHCT in patients with multiple myeloma (*n* = 15) and non-Hodgkin lymphoma (*n* = 6). Six-minute walk test (6MWT; *p* = 0.014), lean mass (*p* = 0.002), and fat mass (*p* = 0.02) decreased; nausea and fatigue increased at FU (both *p* = 0.039). Recent weight change and steroid exposure were predictors of reduced aerobic capacity (*p* < 0.001). There were no significant changes in interleukin (IL)-1β, IL-6, tumor necrosis factor (TNF), energy expenditure, or bioavailable testosterone. Alterations in cytokines, energy expenditure, and testosterone were not associated with functional impairment acutely following AHCT. Recent history of weight loss and steroid exposure were predictors of worse physical function after AHCT, suggesting that targeting nutritional status and myopathy may be viable strategies to mitigate these effects.

## 1. Introduction

Over 33,000 autologous hematopoietic stem cell transplants (AHCT) are performed worldwide annually to treat various hematologic malignancies [1]. Though typically well tolerated with expected short-term toxicities, depending on disease status, AHCT can result in significant morbidity. The direct effects of malignancy, effects of chemotherapy/radiation, drug treatment, and sedentary behavior all likely contribute to increased fatigue and reduced quality of life (QOL) acutely after AHCT [2]. In addition, muscle strength was reportedly lower in patients with hematologic malignancies than age-related healthy controls before initiation of the HCT process [3] likely contributing to reductions in QOL, physical activity, and lean body mass (LBM) after HCT [2,4].

In non-transplant therapy of malignancy, fatigue, muscle wasting, and functional impairment are symptoms of cachexia, and are associated with increases in inflammation and energy expenditure and decreased levels of anabolic hormones such as testosterone. In a patient cohort with solid or hematologic cancer, those with at least 5% weight loss over the prior six months also displayed elevated inflammatory markers, along with reduced LBM, physical function, and bioavailable testosterone [5]. In that same report, C-reactive protein was a significant predictor of LBM and patient-reported fatigue [5]. Additionally, low testosterone has been associated with reduced appetite [6,7], which may further promote fatigue, muscle wasting, and functional impairment through malnutrition. However, the contribution of inflammation, energy expenditure, and testosterone in the AHCT setting has not been characterized.

The purpose of this study was to describe biochemical and physiologic factors that may contribute to acute AHCT-induced muscle wasting and functional impairment. We hypothesized that fatigue, inflammation, and energy expenditure would significantly increase, and testosterone, LBM, and physical function (measured by the six-minute walk test, 6MWT) would significantly decrease, one month after AHCT, and that changes in fatigue, energy expenditure, LBM, and testosterone could be associated with adverse changes in physical function.

## 2. Results

Fifty-one patients were screened, 27 provided consent, 24 participated in baseline (PRE) assessment, and 21 participated in follow-up (FU) assessment (Figure 1). Data are presented as median (95% CI) or N (%) for the 21 patients who participated in both PRE and FU. Baseline demographics, diagnosis, and recent treatment exposure is provided in Table 1. There were 69 (62, 75) days between PRE and FU and 28 (24, 30) days between HCT and FU. Cumulative steroid exposure between PRE and FU was 118 (85.5, 150.5) mg (dexamethasone equivalents); all patients received at least 8 mg. Between PRE and HCT, 13 (61.9%) patients received 3500 (2000, 5000) mg cyclophosphamide only and eight (38.1%) received 6850 (3700, 8600) mg cyclophosphamide plus 1140 (425, 1300) mg etoposide for chemomobilization. For the conditioning regimens, patients with multiple myeloma received 350 (250, 435) mg melphalan and patients with lymphoma received BEAM (carmustine, etoposide, cyratabine, and melphalan) or mini-BEAM regimens: 582.5 (440, 645) mg carmustine, 3080 (1200, 3450) mg cyratabine, 3080 (1200, 3450) mg etoposide, and 272.5 (200, 290) mg melphalan. One patient (4.8%) also received 900 cGy of localized radiotherapy to a foot lesion.

### 2.1. Physical Function

Distance covered in the 6MWT was significantly decreased at FU (Figure 2A); the median change was −40.8 (−85.8, −11.8) m (−8.3% (−15.3%, −2.3%)), which was greater than the minimum clinically important difference of 30.5 m [8]. Other parameters of physical function were also significantly decreased at FU (Table 2): handgrip strength (HGS), stair climb power (SCP), chair stand test (CST), lower body one-repetition maximum (1-RM: Knee Extension, KE; Knee Flexion, KF; Hip Extension, HE), and peak aerobic capacity (VO_2;_
Figure 2B–F; *p* < 0.05). Knee Extension is presented in Figure 2F as a representative example. 

### 2.2. Body Composition and Energy Expenditure

Body weight and appendicular lean mass (ALM) were statistically significantly reduced at FU (Figure 3A,B), as were LBM, appendicular skeletal muscle index (ASMI), and fat mass (FM; Table 2). There were no significant changes from PRE to FU for measured resting energy expenditure (REE; Figure 3C), REE relative to predicted REE (% predicted), REE relative to LBM (Figure 3D), REE relative to ALM, or respiratory quotient (RQ; Table 2).

### 2.3. Patient-Reported Outcomes

Nausea score, as reported by the Anderson Symptom Assessment Scale (ASAS), significantly worsened from PRE to FU (−1.0 (−2.7, −0.1)); *p* = 0.04); there were no significant changes in other ASAS symptom scores. Some categories of the Functional Assessment of Chronic Illness Therapy-Fatigue (FACIT-F) significantly worsened from PRE to FU including social well-being (−1.0 (−3.5, −0.1); *p* = 0.027) and fatigue (−5.0 (−11.3, −0.2); *p* = 0.039, Figure 4A), with a trend for worsened functional well-being (−2.0 (−4.1, 0.2); *p* = 0.064). Vitality score, as reported by Health Survey (SF-36), worsened from PRE to FU (−2.0 (−4.3, −0.1); *p* = 0.039, Figure 4B); there were no significant changes in any other SF-36 categories, or for Positive and Negative Affect Schedule, Global Assessment of Change Questionnaire, Patient Health Questionnaire, Functional Assessment of Cancer Therapy-Bone Marrow Transplant Scale, or Rapid Assessment of Physical Activity scores.

### 2.4. Biomarkers

The proportion of patients with detectable interleukin (IL)-1β levels was not significantly different between PRE (33.1%) and FU (47.6%; *p* = 0.53). There was a trend for circulating tumor necrosis factor (TNF) to increase (*p* = 0.09; Figure 5A) with no significant change in circulating IL-6 from PRE to FU (Figure 5B). There were significant increases in total testosterone (TT) and sex-hormone binding globulin (SHBG), with no significant change in calculated bioavailable testosterone (cBT) from PRE to FU in men (*n* = 19; Figure 5C–E). In men, the proportion of hypogonadism by TT (≤3 ng/mL) was not significantly different between PRE (42.1%) and FU (21.1%; *p* = 0.15). Neither was the proportion of hypogonadism by cBT (≤70 ng/dL) between PRE (10.5%) and FU (26.3%; *p* = 0.21) in men.

### 2.5. Regression

Conditional variables included in multivariate analysis for prediction of 6MWT change or VO_2_ peak change were: age, diagnosis, HCT-Comorbidity Index, number of days to FU, steroid exposure during three months prior to PRE, cumulative dexamethasone exposure between PRE and FU, and PRE values for TT, cBT, IL-6, TNF, six-month relative weight loss, BMI (body mass index), ASMI, 6MWT, VO_2_ peak, SCP, CST, HGS, and 1-RM for KE. Greater relative weight loss six months before PRE (Unstandardized Beta (standard error): B (se) = 0.28 (0.03); *p* < 0.001) and exposure to steroids within three months prior to enrollment in the Bone Marrow Transplant Unit (BMTU) (B (se) = −4.82 (0.32); *p* < 0.001) predicted greater reductions in VO_2_ peak after multivariate analysis. There were no significant predictors for 6MWT change in multivariate regression analysis. The same conditional variables were included in logistic regression as predictors of complications for the top four most frequent complications between PRE and HCT (diarrhea (*n* = 18), nausea/vomiting (*n* = 11), neutropenia (*n* = 10), constipation (*n* = 9)) or between HCT and FU (neutropenia (*n* = 19), diarrhea (*n* = 18), nausea/vomiting (*n* = 16), neutropenic fever (*n* = 11)). There were no statistically significant models for prediction of complication occurrence.

## 3. Discussion

In this study we showed that AHCT in patients with multiple myeloma and non-Hodgkin lymphoma caused decrement in physical function in parallel with reduced lean and fat mass, and increased patient-reported fatigue one month after AHCT. Autologous HCT did not induce significant changes in energy expenditure, inflammatory cytokines, or bioavailable testosterone levels. Even though decline in physical function and quality of life have been previously reported after HCT, the mechanisms underlying these changes are not completely understood. In the setting of non-transplant therapy of malignancy, elevated energy expenditure [9], inflammation [10], and low testosterone [5,7,11] are associated with cachexia and functional impairment, though their relative contribution in the HCT setting is unclear. To our knowledge, this was the first study to report the impact of AHCT on a battery of objective physical function assessments in combination with energy expenditure, inflammatory cytokines, testosterone, and patient-reported QOL outcomes following AHCT. This was also the first assessment of the impact of AHCT on physical function and QOL in U.S. Veterans, a population with a higher comorbidity burden and increased risk for hematologic malignancies than the general population [12,13].

The effects of AHCT on physical performance as measured by the 6MWT are not well-established. Whereas a significant reduction in 6MWT was reported acutely after allogeneic HCT [14,15], no change was reported after AHCT in a different report [16]. In other disease populations, change in 6MWT is clinically relevant. A systematic review comprised of studies including patients with chronic obstructive pulmonary disease, lung cancer, coronary artery disease, other lung diseases, or adults with fear of falling, reported that the minimal clinically important difference, determined with ≥70% specificity via receiver operating characteristic curve analysis, for a patient to perceive a change in 6MWT performance, was between 14–30.5 m [8]. While this criterion has not been validated in hematologic malignancy populations, this range is markedly smaller than the 40.8 m median decrease observed in the current study, suggesting that, while most patients performed ≥400 m prior to AHCT (the low end of the normal range reported in healthy individuals) [17], the negative impact on functional capacity could be significant. 

We also observed significant reductions in other functional parameters, indicating a broad impact on overall physical fitness. VO_2_ peak, CST, lower body 1-RM, SCP, and HGS all decreased one-month after HCT, indicating a reduction in aerobic capacity, muscle endurance, muscle strength, muscle power, and overall functional performance, respectively. Functional decline using different tools has been shown before [18,19]; however, our observations were important because they provided evidence for the feasibility of performing a battery of functional performance tests before and shortly after AHCT.

As there is no validated tool to assess physical function in patients with malignancies, this study provided data on the performance of different functional tests in a subset of this population. The functional tests we reported here have been previously validated in other settings and were well tolerated. However, some such as VO_2_ peak may not be feasible in larger multicenter clinical trials or general clinical practice. Tests such as 1-RM require equipment that are not easily accessible outside of research settings. Nevertheless, others including 6MWT, HGS, SCP, and CST, may be more feasible options and provide clinicians with inexpensive, easy to perform tools to assess physical function. 

We observed a significant and proportionate reduction in LBM and FM, as evidenced by a lack of change in body fat percentage. Also, there were greater relative decreases in all functional parameters than the decreases in LBM or ALM, indicating a reduction in muscle quality as well as muscle mass. Compared to age-matched healthy controls, lower limb muscle quality was reduced in male and female gastrointestinal cancer patients with weight loss [20] but not in Hodgkin’s lymphoma survivors when assessed five years after diagnosis [21]. The early impact of AHCT on muscle quality is unknown; however, our observations provided the first evidence of short-term reductions in muscle quality after AHCT. The clinical relevance of muscle quality assessment lies in the dissociation between improvements in muscle mass and lack of improvement in muscle quality and/or physical function in interventions aimed at improving muscle wasting and functional impairment in the cancer setting [22,23].

To our knowledge, this was the first report of the effects of AHCT on REE in adults. Increases in REE have been shown in the setting of cancer in some [9] but not in all studies [24] and this is postulated as one mechanism contributing to muscle wasting. Reductions in mass, particularly LBM, without changes in REE, may imply an increase in relative energy expenditure. However, this was not observed in the present study even after adjusting for LBM. This finding may indicate that the reduction in muscle mass was insufficient to impact REE; however, skeletal muscle comprises 20%–30% of REE in adults, and other lean organs could have increased metabolic demand [25]. 

Previous studies suggest that cytokine levels peak near white blood cell nadirs, around one-week post AHCT, and return to pre-AHCT levels one month later [26,27]. These studies report that peak levels of IL-6 coincides with peak symptom levels of pain, fatigue, anorexia, sleeplessness, and drowsiness [26,27]; however, these studies do not evaluate the association between inflammation and energy expenditure, body weight, body composition, or objective physical function. In our study, there were no significant changes in inflammatory cytokine levels one-month after AHCT. Given that energy expenditure is thought to be modulated by inflammation [28], this is consistent with the lack of effect of AHCT on REE we reported here. Inflammation is also proposed as one of the mechanisms mediating fatigue and muscle weakness in the setting of cancer [29] and the fact that these outcomes worsened significantly in spite of inflammatory cytokines remaining stable suggests that other mechanisms may contribute to the pathogenesis of these symptoms. One factor specific to this population that could explain this lack of effect is the use of glucocorticoids that are known to induce muscle weakness despite their anti-inflammatory properties [30]. This may partially explain the observation of exposure to steroids within three months prior to enrollment in the BMTU as a predictor of reduced aerobic capacity. Larger studies will be needed to answer these questions. 

Bioavailable testosterone is the portion of total testosterone that circulates unbound to SHBG and is available for binding to the androgen receptor in target organs [31]. It is suggested that assessment of cBT or free T (unbound to SHBG or albumin) is more appropriate than that of TT alone for diagnosing hypogonadism, particularly in the cancer setting [5]. In the current study, circulating TT levels increased without changes in cBT due to the increase in SHBG. As a result, the proportion of patients with low T was unchanged one month after AHCT. This observation contrasts with reports that TT transiently decreases in the first few weeks-to-months after AHCT with low TT in one third of patients one year following AHCT [32]. To our knowledge, this was the first report of one-month changes in TT, cBT, and SHBG in a cohort of AHCT recipients. A case report of a patient with sickle-cell disease who underwent allogeneic HCT reported an increase in TT one month after HCT [33]; however, they also reported a decrease in SHBG and calculated free T which were not observed in the current study. This data promotes the relevance of BT assessment in addition to TT when HCT patients have symptoms consistent with hypogonadism. However, the efficacy of testosterone replacement was not proven in this setting and a clinical trial would be needed before making broad recommendations. 

This study has several strengths including the homogeneity of the population, a comprehensive assessment of physical function using clinically validated tools, measurements of testosterone using the gold standard method liquid-chromatography tandem mass spectroscopy (LC/MSMS) and simultaneously assessing the patients’ perspective using patient-reported outcomes (PRO). The changes in physical function after AHCT we demonstrated in this report using different tools could also provide guidance in the selection of outcomes for future clinical trials targeting muscle wasting, cachexia, or fatigue in patients undergoing AHCT. Some important limitations should be noted for this pilot study such as the relatively small sample size, with larger studies required to verify these observations. We reported a single follow-up time-point very early after HCT and future studies should aim to determine whether these acute observations are associated with long-term clinical outcomes. This will be important considering the recent calls for assessment of cardiovascular and/or metabolic syndrome risk factors after HCT [34,35,36]. In addition, we did not assess nutritional status or spontaneous physical activity via accelerometry. 

## 4. Materials and Methods

### 4.1. Study Participants

Patients with confirmed myeloma or lymphoma planning treatment with AHCT at the Veterans Affairs Puget Sound Health Care System (VAPSHCS) BMTU in Seattle, WA, USA were eligible. Patients who declined participation were excluded. The protocol was approved by the VAPSHCS Institutional Review Board (human studies protocol #00935) and the Research and Development Committee and was conducted in compliance with the Declarations of Helsinki and its amendments and the International Conference on Harmonization Guideline for Good Clinical Practices. Recruitment took place between February 2017 and July 2018.

### 4.2. Study Design and Protocol

Study measurements were assessed after enrollment into the BMTU, occurring prior to chemomobilization (PRE), and again 30 ± 10 days after HCT (FU; Figure 1). For PRE and FU visits, patients reported to VAPSHCS in the morning after fasting overnight. A blood sample was obtained before 10 AM to measure circulating inflammatory and hormone markers followed by assessment of body composition, resting energy expenditure (REE), physical function, and patient-reported outcomes (PRO).

Physical function was assessed by 6MWT (primary outcome); VO_2_ peak (Vmax Encore, Vyaire Medical, Inc., Mettawa, IL, USA.); HGS (Jamar Hydraulic Dynamometer, J.A. Preston Corp., Clifton, NJ, USA); SCP; CST; lower body 1-RM (knee extension (KE), knee flexion (KF), hip extension (HE); Kaiser Sports Health Equipment, Inc., Fresno, CA, USA)). For 6MWT, participants walked back and forth for six continuous minutes down a hallway of 30 meters and the total distance covered was recorded. VO_2_ peak was assessed via indirect calorimetry while wearing a mask with a breathing valve to collect expired gases (Vmax Encore, Vyaire Medical, Inc.). Participants pedaled a cycle ergometer at progressively harder workloads, while maintaining 55–65 rpm speed, until volitional fatigue or until the patient could not maintain the required speed. The participant’s heart rate was monitored by a chest strap attached to a heart rate monitor. Participants were asked to rate their perceived exertion using the Borg Scale [37]. HGS was measured by a handheld dynamometer (Jamar Hydraulic Dynamometer, J.A. Preston Corp.). To assess SCP, participants climbed a flight of standard hospital stairs (13 steps, 15.3 cm each) at the highest possible speed, according to their capabilities [38]. Two–three trials were attempted, and the shortest time employed to complete the test was recorded. Muscle strength was measured by 1-RM of the major muscle groups of the lower body (knee extension, knee flexion, right hip extension) following American College of Sports Medicine strength testing guidelines [39]. The CST determined the number of times a patient could stand up and sit down in 30 seconds from a fully seated position in a chair with arms crossed across the chest [40]. Two–three trials were attempted and the highest number of repetitions in a single trial was recorded. 

Body composition was measured by dual-energy X-ray absorptiometry (Hologic Inc., Marlborough, MA, USA) for LBM, ALM, FM, percent body fat (PBF), and ASMI Equation (1) [41]. REE was assessed by indirect calorimetry. Patients rested in a supine position before assessment using a ventilated hood technique (Vmax Encore, Vyaire Medical, Inc.). The measurements collected in the last 20 min were averaged to calculate REE (kcals/d) and respiratory quotient (VCO_2_/VO_2_ ratio; RQ) as a measure of relative utilization of carbohydrates and fat [42].
(1)$$\mathrm{ASMI}\;=\;\left(\mathrm{ALM}\;\mathrm{kg}\right)/\mathrm{height}\;(m^{2})$$

Patient-reported outcomes included: Anderson Symptom Assessment Scale (ASAS) [43], Functional Assessment of Chronic Illness Therapy-Fatigue (FACIT-F) [44]. Short Form (36) Health Survey [45], Positive and Negative Affect Schedule [46], Global Assessment of Change Questionnaire [47], Patient Health Questionnaire [48], Functional Assessment of Cancer Therapy-Bone Marrow Transplant Scale [49], and Rapid Assessment of Physical Activity [50]. All scores were converted as necessary so that larger numbers and/or positive change scores indicated better/improved QOL and smaller numbers and/or negative change scores indicated worse/reduced QOL.

Inflammatory cytokines in plasma were detected by V-PLEX Human Pro-inflammatory Panel 1 from Meso Scale Discovery company (Cat# N05049 A-1, MSD, Rockville, MD, USA). The V-PLEX kit contains IL-1β, IL-6, and TNF spotted in each well as sandwich immunoassays. A protocol provided by the manufacturer was used for this assay. In brief, the plasma samples and controls were diluted in two-fold before adding to the 96-well V-PLEX plate. The plate was incubated at room temperature with shaking for two hours followed by three washes in phosphate buffered saline with 0.05% Tween 20 (PBS/T). After adding 1× detection antibody solution, the plate was incubated at room temperature with shaking for another two hours. After another three washes in PBS/T, the Read Buffer T containing electrochemiluminescent labelled (SULFO-TAG) detection antibody was added. The V-PLEX plate was read on Meso Scale Discovery (MSD) Sector Imager (Rockville) and the data was analyzed by Discovery Workbench v4.0. software (Rockville). Total testosterone and SHBG in plasma were analyzed via LC/MSMS as previously reported [51]. Bioavailable testosterone was calculated with theoretical association constants for testosterone with SHBG and albumin [52].

### 4.3. Statistical Analysis

SPSS v18 (SPSS, Inc., Chicago, IL, USA) was used for statistical analysis. The primary endpoint of this study was the change in 6MWT distance, a useful tool for assessment of exercise capacity [53]; all other measures were considered exploratory. Comparison between PRE and FU were analyzed using Wilcoxon signed rank test. Statistical significance was two-sided, α ≤ 0.05. Multiple regression controlling for collinearity was used to determine baseline predictors of change for 6MWT or aerobic capacity; logistic regression was used to determine baseline predictors of complication occurrence. Statistical comparison between PRE and FU for categorical data were performed using the Fisher’s exact test or Chi-squared test.

## 5. Conclusions

Autologous HCT led to acute adverse effects on physical function and QOL by inducing deconditioning, fatigue, and muscle atrophy. In this setting, these observations were seen despite unchanged REE, inflammatory cytokines, and testosterone levels, suggesting that other mechanisms were at play. These observations are clinically relevant given that treatment strategies currently in development for cachexia in cancer include anti-cytokines [54], androgens [55], and agents that may decrease energy expenditure [56]. Our data suggest that these strategies may not fully address the alterations leading to muscle wasting and functional impairment acutely after AHCT. The fact that weight loss history prior to AHCT was a significant predictor of worse physical function after transplantation suggests that targeting nutritional status may be a viable strategy to prevent this complication. Future studies should also aim to elucidate the prognostic value of functional performance measures on long-term outcomes such as infection, hospitalization, relapse, and survival.

## Figures and Tables

**Figure 1 cancers-11-01300-f001:**
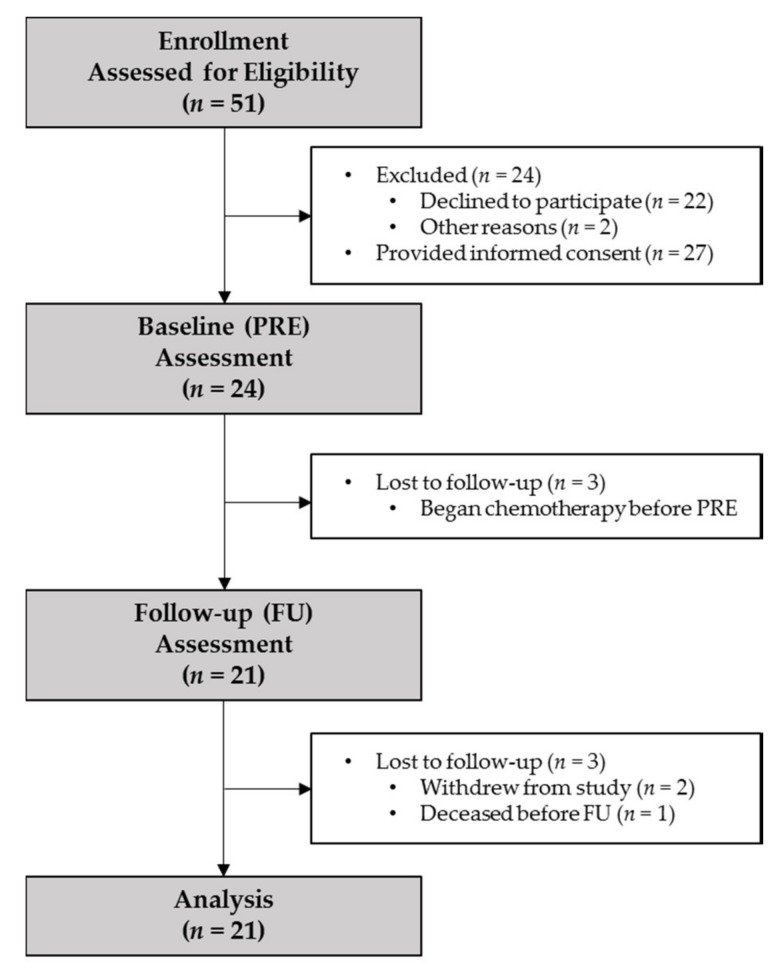
Study flow diagram. PRE, baseline visit; FU, one-month follow-up visit.

**Figure 2 cancers-11-01300-f002:**
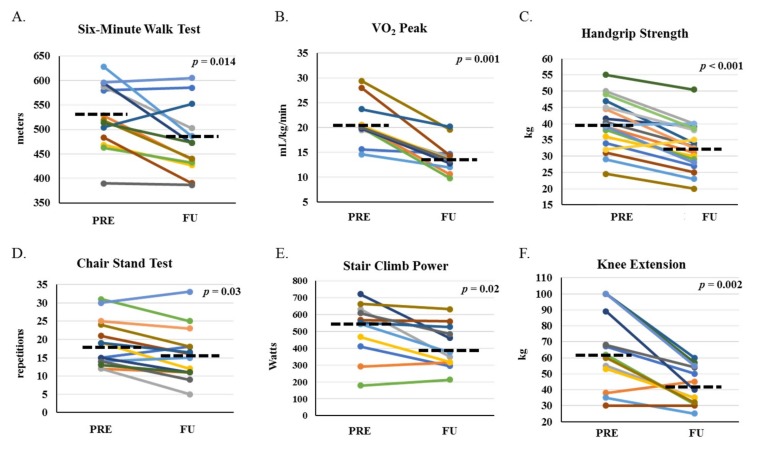
Physical function before and one-month after HCT. Physical function performance at PRE and FU for six-minute walk test (**A**), aerobic capacity (VO_2_ peak) (**B**), average handgrip of two individual hands (**C**), chair stand test (**D**), stair climb power (**E**), and knee extension one-repetition maximum as a representative image of lower body strength (**F**). *p*-values indicate paired-sample comparison between PRE and FU. Dashed lined represent median at PRE and FU.

**Figure 3 cancers-11-01300-f003:**
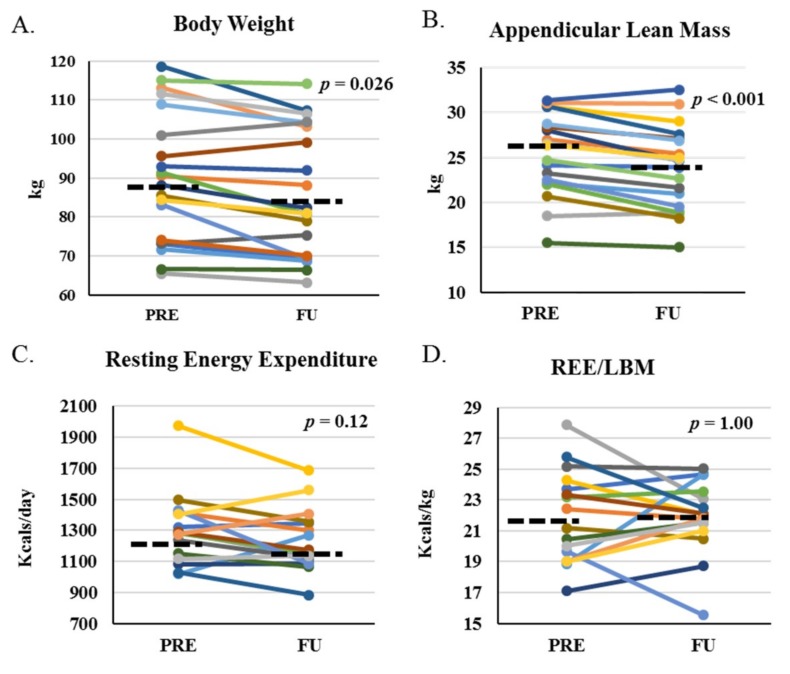
Body composition and energy expenditure before and one-month after HCT. Body composition measures for total body weight (**A**), appendicular lean mass (**B**), resting energy expenditure as measured (**C**), and relative to total lean mass (**D**) at PRE and FU. *p*-values indicate paired-sample comparison between PRE and FU. Dashed lined represent median at PRE and FU.

**Figure 4 cancers-11-01300-f004:**
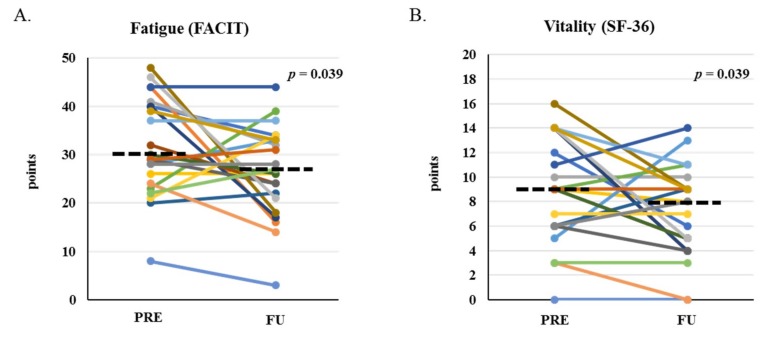
Patient-reported fatigue categories before and one-month after HCT. Patient-reported fatigue as assessed by the fatigue sub-section of Functional Assessment of Chronic Illness Therapy-Fatigue (FACIT-F) (**A**) and vitality sub-section of Health Survey SF-36 (**B**) at PRE and FU. *p*-values indicate paired-sample comparison between PRE and FU. Dashed lined represent median at PRE and FU.

**Figure 5 cancers-11-01300-f005:**
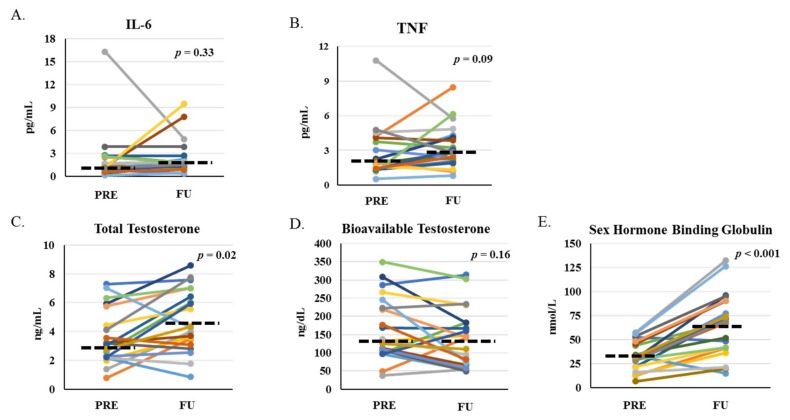
Biomarkers before and one-month after HCT. Circulating levels of inflammatory cytokines interleukin (IL)-6 (**A**) and tumor necrosis factor (TNF) (**B**) for men and women and anabolic markers: total testosterone (**C**) and calculated bioavailable testosterone (**D**) for men only, and sex-hormone binding globulin (**E**) for men only at PRE and FU. *p*-values indicate paired-sample comparison between PRE and FU. Dashed lined represent median at PRE and FU.

**Table 1 cancers-11-01300-t001:** Baseline characteristics.

*n* = 21	Med (95% CI)
Age (yr)	68 (57, 68)
Ht (cm)	174.6 (170.4, 178.0)
Wt (kg)	88.7 (83.6, 100.7)
BMI (kg/m^2^)	28.3 (27.8, 32.3)
ASMI (kg/m^2^)	8.6 (8.0, 9.1)
HCT-CI range (0–7) ^a^	3.0 (1.9, 3.9)
	N (%)
Men	19 (90.5)
Ethnicity	
White Non-Hispanic	13 (61.9)
White Hispanic	1 (4.8)
Black	3 (14.3)
Asian/Pacific Islander	2 (9.5)
Mixed	2 (9.5)
Diagnosis	
Multiple Myeloma	15 (71.4)
T-cell Lymphoma	1 (4.8)
Hodgkin Lymphoma	1 (4.8)
B-cell non-Hodgkin Lymphoma	4 (19.0)
Recent chemotherapy exposure (y) ^b^	
Alkylating Agents	2 (9.5)
Topoisomerase Inhibitors	6 (28.6)
Immunomodulators	6 (28.6)
Proteasome Inhibitors	7 (33.3)
None	10 (47.6)
Glucocorticoid exposure (y) ^c^	10 (47.6)

^a^ Range for this cohort; ^b^ within one month prior to enrollment in Bone Marrow Transplant Unit (BMTU); ^c^ within three months prior to enrollment in BMTU; BMI, body mass index; ASMI, appendicular skeletal mass index; HCT-CI, hematopoietic cell transplant comorbidity index; y, yes.

**Table 2 cancers-11-01300-t002:** Changes in physical function, body composition, energy expenditure, and biomarkers.

Med (95% CI)	Absolute Change	Relative Change	*p*-Value ^a^
**Physical Function**			
6MWT (m)	−40.8 (−85.8, −11.8)	−8.3 (−15.3, −2.3)	0.014
HGS (kg)	−6.5 (−8.2, −3.5)	−16.7 (−20.2, −8.6)	0.001
SCP (W)	−73.7 (−186.0, −19.7)	−17.1 (−31.6, −0.8)	0.024
CST (reps)	−3.0 (−5.1, −0.4)	−17.4 (−29.1, −1.7)	0.025
KE (kg)	−20.0 (−3.5, −10.9)	−36.4 (−46.4, −16.4)	0.002
KF (kg)	−15.0 (−21.4, −5.0)	−22.4 (−31.4, −7.6)	0.005
HE (kg)	−4.5 (−11.8, −0.4)	−13.0 (−29.7, −3.7)	0.038
VO_2_ peak (mL/kg/min)	−6.7 (−8.8, −3.6)	−33.3 (−41.0, −18.1)	0.001
**Body Composition**			
BW (kg)	−3.9 (−6.3, −2.0)	−4.2 (−6.6, −2.2)	0.001
LBM (kg)	−2.9 (−4.0, −1.0)	−4.9 (−6.7, −1.6)	0.002
ALM (kg)	−1.6 (−2.1, −0.9)	−6.1 (−8.6, −3.6)	<0.001
ASMI (kg/m^2^)	−0.5 (−0.7, −0.3)	−6.0 (−8.7, −3.6)	0.001
FM (kg)	−0.7 (−1.5, −0.1)	−2.3 (−7.1, −0.4)	0.02
PBF (%)	0.5 (−0.6, 0.8)	n/a	0.78
**Energy Expenditure**			
REE (kcals/d)	−113 (−148,18)	−8.4 (−11, 2)	0.12
REE (% predicted)	−1.5 (−6.1, 3.9)	n/a	0.64
RQ (VCO_2_/VO_2_)	−0.04 (−0.08, 0.02)	−4.1 (−8.5, 2.5)	0.22
REE/LBM (kcals/kg)	0.1 (−1.5, 1.4)	0.5 (−6.1, 7.6)	1.00
REE/ALM (kcals/kg)	1.6 (−2.9, 4.1)	2.8 (−4.8, 9.4)	0.56
**Biomarkers**			
IL−6 (pg/mL)	0.26 (−1.67, 1.61)	13.7 (−12.3, 154.2)	0.33
TNF (pg/mL)	0.65 (−0.37, 1.71)	52.4 (8.0, 109.0)	0.09
TT (ng/mL) ^b^	1.3 (0.3, 1.9)	25.3 (19.5, 107.5)	0.02
cBT (ng/dL) ^b^	−35.4 (−51.2, 8.1)	−13.3 (−26.9, 29.6)	0.16
SHBG (nmol/L) ^b^	41.2 (22.8, 44.1)	122.8 (79.3, 145.9)	<0.001

^a^*p*-values represent paired-sample *t*-test between PRE and FU; ^b^ men only (*n* = 19); 6MWT, six-minute walk test; HGS, handgrip strength, CST, chair stand test; SCP, stair climb power; KE, knee extension; KF, knee flexion; HE, hip extension; CP, chest press; LP, latissimus pull-down; UB, upper back row; VO_2_, volume of oxygen; BW, body weight; LBM, lean body mass; ALM, appendicular lean mass; ASMI, appendicular skeletal mass index; FM, fat mass; PBF, percent body fat; REE, resting energy expenditure; RQ, respiratory quotient; VCO_2_, volume of carbon dioxide; VO_2_, volume of oxygen; IL, interleukin; TNF, tumor necrosis factor; TT, total testosterone; cBT, calculated bioavailable testosterone; SHBG, sex-hormone binding globulin.

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
