# Peer review of "Assessing Cachexia Acutely after Autologous Stem Cell Transplant"

_cancers, 2019, doi:10.3390/cancers11091300_

Round 1

Reviewer 1 Report

This manuscript by Anderson et al investigates acute effects of autologous stem cell transplant on physical function, body composition, QOL, energy expenditure and androgens. Strengths include that there is a pre/post test analysis to test acute changes that occur after transplant as well as the types of measures collected including body mass and the 6-minute walk test--an important test for physical function.  Findings of the study are quite important showing a decrease in physical function, lean mass and fat mass acutely.  This makes the important argument that pre-habilitation prior to transplant may be helpful and worthy investigating in the future.

Weaknesses include: 1. This is an observational study and not randomized, however, this manuscript is one of the first to describe these changes so this remains highly worthy of publication.

2. There is no information on how long patients were on chemotherapy for their cancer. Although it is reported which chemotherapies were used as conditioning agents prior to transplant, there is no info on total chemo exposure or number of lines of therapy prior to transplant. I think the reader would like to see it.  It might give some context on why certain subjects had more weight loss prior to transplant and possibly why these subjects had worse outcomes.

3. Would be very interesting to see what happened to these subjects 3 months or 1 year post transplant, albeit it is harder to capture these measures in this type of study.  If obtaining this information is available, it could make the manuscript stronger, but manuscript currently very strong on its own.

Author Response

Reviewer 1 comments and author responses:

This manuscript by Anderson et al investigates acute effects of autologous stem cell transplant on physical function, body composition, QOL, energy expenditure and androgens. Strengths include that there is a pre/post-test analysis to test acute changes that occur after transplant as well as the types of measures collected including body mass and the 6-minute walk test--an important test for physical function.  Findings of the study are quite important showing a decrease in physical function, lean mass and fat mass acutely.  This makes the important argument that pre-habilitation prior to transplant may be helpful and worthy investigating in the future.

Weaknesses include: 1. This is an observational study and not randomized, however, this manuscript is one of the first to describe these changes so this remains highly worthy of publication.

Author Response: We agree that randomization is necessary in future trial design but greatly appreciate this reviewers’ comments that lack of randomization is acceptable in this preliminary observational report.

There is no information on how long patients were on chemotherapy for their cancer. Although it is reported which chemotherapies were used as conditioning agents prior to transplant, there is no info on total chemo exposure or number of lines of therapy prior to transplant. I think the reader would like to see it. It might give some context on why certain subjects had more weight loss prior to transplant and possibly why these subjects had worse outcomes.

Author Response: The authors appreciate this recommendation and have now added the total chemotherapy exposure each for mobilization and conditioning regimens in Lines 65-70.

Would be very interesting to see what happened to these subjects 3 months or 1 year post transplant, albeit it is harder to capture these measures in this type of study. If obtaining this information is available, it could make the manuscript stronger, but manuscript currently very strong on its own.

Author Response: We agree that the logical progression from this report will be to assess the relationship between these acute changes and long-term follow up and greatly appreciate this reviewers’ comments of the current strength of the manuscript.

Reviewer 2 Report

In this manuscript by Anderson et al, factors associated with cancer cachexia were assessed in 21 patients after autologous HCT. The observed acute adverse effects on physical function and QOL, demonstrated by deconditioning, fatigue, and muscle atrophy at about 1 month post transplant. However, there was no significant change in REE, inflammatory cytokines, and testosterone levels.

Comments:

1)   The authors should be congratulated in prospectively evaluating cachexia in auto-HCT recipients, as in general this not well understood and of importance for this patient population. The manuscript is well organized and well written.

2)   My biggest concern is that the follow up time point is still very early after transplant. Mortality with auto-HCT is very low. And while it is true that patients experience acute side effects and impairment in QoL early after transplant, the general experience is that by 2-3 months post-transplant patients are clinically close to their baseline. What would the longitudinal change look like in patients at 3 months after transplant (i.e. baseline vs 1 month post vs 3 months post)? Any abnormalities persistent or all resolved? While I agree with the authors that the investigation into these cachexia-related factors is important, the limitation in the follow up time period should be further emphasized.

There is overlap in cancer cachexia and body composition changes, which also impacts cardiovascular health. There is a lot on interest in better understanding these factors in the long-term health of transplant recipients, including understanding longitudinal changes in body composition (PMID: 29496561). There have been recent guidelines recommendations from the NIH (PMID: 27590105) and the ASBMT/EBMT (PMID: 27184625) emphasizing the need to further investigate these factors (such as this study) in transplant recipients. Consider making reference to these initiates in the context of your work.

Please report the chemotherapy condition regimens that the patients received.

Author Response

Reviewer 2 comments and author responses:

In this manuscript by Anderson et al, factors associated with cancer cachexia were assessed in 21 patients after autologous HCT. The observed acute adverse effects on physical function and QOL, demonstrated by deconditioning, fatigue, and muscle atrophy at about 1 month post-transplant. However, there was no significant change in REE, inflammatory cytokines, and testosterone levels.

Comments:

1)   The authors should be congratulated in prospectively evaluating cachexia in auto-HCT recipients, as in general this not well understood and of importance for this patient population. The manuscript is well organized and well written.

Author Response: The authors greatly appreciate this acknowledgement of the relevance for this patient population

2)   My biggest concern is that the follow up time point is still very early after transplant. Mortality with auto-HCT is very low. And while it is true that patients experience acute side effects and impairment in QoL early after transplant, the general experience is that by 2-3 months post-transplant patients are clinically close to their baseline. What would the longitudinal change look like in patients at 3 months after transplant (i.e. baseline vs 1 month post vs 3 months post)? Any abnormalities persistent or all resolved? While I agree with the authors that the investigation into these cachexia-related factors is important, the limitation in the follow up time-period should be further emphasized. 

There is overlap in cancer cachexia and body composition changes, which also impacts cardiovascular health. There is a lot on interest in better understanding these factors in the long-term health of transplant recipients, including understanding longitudinal changes in body composition (PMID: 29496561). There have been recent guidelines recommendations from the NIH (PMID: 27590105) and the ASBMT/EBMT (PMID: 27184625) emphasizing the need to further investigate these factors (such as this study) in transplant recipients. Consider making reference to these initiates in the context of your work.

Author Response: The authors agree that the logical progression from this report will be to assess the relationship between these acute changes and long-term follow up and that these references will provide an important context for readers. These citations have been added along with the following statement in Lines 253-255 “This will be important considering the recent calls for assessment of cardiovascular and/or metabolic syndrome risk factors after HCT.” The follow-up time-period has also been emphasized as a limitation in the discussion in Lines 252-253.

Please report the chemotherapy condition regimens that the patients received.

Author Response: The authors appreciate this recommendation and have now added the total chemotherapy exposure each for mobilization and conditioning regimens in Lines 65-70.